# The Use of Mushrooms and Spirulina Algae as Supplements to Prevent Growth Inhibition in a Pre-Clinical Model for an Unbalanced Diet

**DOI:** 10.3390/nu13124316

**Published:** 2021-11-29

**Authors:** Roni Sides, Shelley Griess-Fishheimer, Janna Zaretsky, Astar Shitrit, Rotem Kalev-Altman, Reut Rozner, Olga Beresh, Maïtena Dumont, Svetlana Penn, Ron Shahar, Efrat Monsonego-Ornan

**Affiliations:** 1Institute of Biochemistry and Nutrition, The Robert H. Smith Faculty of Agriculture, Food and Environment, The Hebrew University of Jerusalem, Rehovot 7610001, Israel; ronisides536@gmail.com (R.S.); shelley.gg@gmail.com (S.G.-F.); janna444@gmail.com (J.Z.); astar.shitrit@mail.huji.ac.il (A.S.); rotem.kalev@mail.huji.ac.il (R.K.-A.); reut.rozner@mail.huji.ac.il (R.R.); olga.andryushche@mail.huji.ac.il (O.B.); s9473net@yahoo.com (S.P.); 2Koret School of Veterinary, The Robert H. Smith Faculty of Agriculture, Food and Environment, The Hebrew University of Jerusalem, Rehovot 7610001, Israel; maitena.dumont@gmail.com (M.D.); ron.shahar1@mail.huji.ac.il (R.S.)

**Keywords:** unbalanced diet, *Spirulina* algae, *Pleurotus eryngii*, *Agaricus bisporus*, microbiota, bone quality

## Abstract

Today’s eating patterns are characterized by the consumption of unbalanced diets (UBDs) resulting in a variety of health consequences on the one hand, and the consumption of dietary supplements in order to achieve overall health and wellness on the other. Balanced nutrition is especially crucial during childhood and adolescence as these time periods are characterized by rapid growth and development of the skeleton. We show the harmful effect of UBD on longitudinal bone growth, trabecular and cortical bone micro-architecture and bone mineral density; which were analyzed by micro-CT scanning. Three point bending tests demonstrate the negative effect of the diet on the mechanical properties of the bone material as well. Addition of *Spirulina* algae or *Pleurotus eryngii* or *Agaricus bisporus* mushrooms, to the UBD, was able to improve growth and impaired properties of the bone. 16SrRNA Sequencing identified dysbiosis in the UBD rats’ microbiota, with high levels of pro-inflammatory associated bacteria and low levels of bacteria associated with fermentation processes and bone related mechanisms. These results provide insight into the connection between diet, the skeletal system and the gut microbiota, and reveal the positive impact of three chosen dietary supplements on bone development and quality presumably through the microbiome composition.

## 1. Introduction

Malnutrition refers to deficiencies (undernutrition), excesses (over-nutrition), or imbalances in a person’s intake of energy and/or nutrients [1]. Micro-nutrient deficiency is another form of malnutrition which until recently was only associated with undernutrition, however today it is well established as a deficiency that can also happen in overweight and obese people. This phenomenon occurs due to consumption of energy-dense, micronutrient-poor foods and drinks and named: the hidden hunger [2].

The popular diets produced and consumed for the past 50 years are no longer nutritionally optimal; they typically contain high amounts of calories, sugar, fat and additives and are depleted in dietary fibers, essential amino acids and micronutrients [3]. This kind of eating pattern, is nowadays common also in low- and middle-income countries, which are also suffering from hunger conditions [4].

The deleterious effects of unbalanced diet (UBD) consumption are well established, and include obesity, dyslipidemia, heart disease, high blood pressure, and impact on daily performance, behavior and mood [5,6]. Yet, there are not enough studies discussing the effect of such a diet on skeletal development and quality [7,8]. Many factors including genes, ethnicity, gender, hormones, physical activity level and diet have a significant impact on bone development, quality, and peak bone mass (PBM) achievement with sexual maturation [9]. Twenty to forty percent of the variance in PBM is determined by lifestyle choices; thus, although heredity is the major contributor, optimization of lifestyle factors is an important strategy to maintain bone health in later life [9].

The effect of diet on skeletal tissue and bone health may be mediated through the microbial communities. The gut microbiota (GM) is acquired at birth and can be modulated by a number of environmental factors such as antibiotic treatments, physiologic state, diseases and of course diet [10]. Alterations in the microbiome have already been associated with variety of conditions, such as inflammatory bowel diseases (IBD), obesity, cardiovascular disease [10,11], and bone health [12]. So far, four potential mechanisms have been suggested as potential linkers between the GM and the skeleton: (1) nutrient absorption and vitamins biosynthesis, (2) the immune system, (3) microbe associated molecular patterns (MAMPs) across the endothelial barrier and (4) signaling molecules/Hormones alteration [13].

In addition to the increased consumption of UBD, the consumption of dietary supplements is in a constant rise as well; according to the Council for Responsible Nutrition report from 2019, 77% of Americans, more than 170 million people, informing they consume dietary supplements [14]. People are interested in easy and natural ways to improve their health status. Among the dietary supplements are vitamins, minerals, protein, amino acids, herbs and botanical [14,15].

*Spirulina* algae (SP) are filamentous cyanobacteria belonging to the *Oscillactoriaceae* family. SP is mainly known by its high content of protein, providing all the essential amino acids with a great bioavailability. Additionally, two essential fatty acids are found in the algae: linoleic (18:2 n-6) and γ-linolenic (18:3 n-6), which are very limited in the common diet. SP is also considered as good source for a variety of vitamins, minerals and functional ingredients such as phycocyanin, β-carotene and polysaccharides [16,17,18]. 

Nutrient composition is relatively similar in different mushrooms. *Pleurotus eryngii* (PE) and *Agaricus bisporus* (AB), commonly known as the king oyster mushroom and the white button mushroom, respectively, are two of the most widely cultivated mushrooms [19]. They are low in fat, containing protein rich in essential amino acids and fibers. Polysaccharides are the main part of the carbohydrate fraction of the mushrooms and have a variety of beneficial health properties [19,20]. Mushrooms are also good sources of several vitamins, particularly from the B complex and minerals as phosphorus, potassium and magnesium. The bioactive substances found in mushrooms are polysaccharides, mainly β-glucans and secondary metabolites as polyphenols [20,21]. 

Lowering blood glucose, improving plasma lipid concentration and antioxidant activity are just a few of the health benefits associated with algae and medicinal mushrooms [22,23,24,25]. In spite of the rising popularity and the intensive research of these supplements, the impact on bone development and quality has not been investigated yet.

In the present study, we established a model that mimics the modern UBD and tested its effects on the growth and skeletal development in young rats. Moreover, we evaluated the impact of botanical dietary supplements, *Spirulina* algae and *Pleurotus eryngii* and *Agaricus bisporus* mushrooms when added to this diet. 

## 2. Materials and Methods

### 2.1. Animals

Forty-eight three weeks old Female Sprague Dawley (SD) post weaning rats, were purchased from Harlan Laboratories (Rehovot, Israel) and housed under environmentally controlled conditions. All procedures were approved by the Hebrew University Animal Care Committee (permit number AG-18-15530-2). After 5 days of adaptation to control chow diet, we randomly divided the rats into six groups, with eight rats in each group: (1) Control (CON); (2) Unbalanced diet (UBD); (3) Unbalanced diet + *Spirulina* algae (UBD + SP); (4) Unbalanced diet + *Pleurotus eryngii* mushroom (UBD + PE); (5) Unbalanced diet + *Agaricus bisporus* mushroom (UBD + AB); and (6) Unbalanced diet + SP + PE + AB (UBD + MIX). At the end of the experiment, after 6 weeks, the rats were anesthetized with isoflurane and blood and caecum samples were collected and stored at −80 °C. Next, their femora, tibias, and vertebra were harvested. The femora and lumbar vertebrae were manually cleaned of soft tissue, wrapped in saline soaked gauze and stored at −20 °C until analysis (mechanical/micro-CT testing). Tibiae were fixed immediately after sacrifice for histological analysis.

### 2.2. Diet Preparation and Composition 

During the postweaning period rats from the CON group were fed a diet based on the recommendation of the American Institute of Nutrition (AIN-93G) that was formulated for the growth phase of rodents: 16% fat, 63.5% carbohydrate and 20.5% protein with the addition of multi vitamins and multi minerals based on AIN-93G as well (Table 1) [26]. The UBD groups were fed a diet high in fat and low in protein, vitamins and minerals, to mimic the popular modern unbalanced diet: 25% fat, 65% carbohydrate, and 10% protein with only 50% of the AIN-93G recommendation for vitamins and minerals for rodents in growth phase (Table 2). The entire diet was homogenized, shaped as dumplings and frozen at −20 °C. The UBD group was the basis for the supplemented diets, with either *Spirulina* algae, *Pleurotus eryngii* mushroom, *Agaricus bisporus* mushroom or all three. The supplements were added to the UBD as dry powder proportionally to the daily intake recommendations for humans (Table 3). The UBD + MIX diet contain the combined dosage of all three supplements. 

### 2.3. Histological Staining of Growth-Plate (GP) Sections

Safranin-O staining was used to examine the tibial GPs. Tibia samples were fixed overnight in 4% paraformaldehyde (PFA, Sigma, St Louis, MO, USA) at 4 °C followed by 3 weeks of decalcification in 0.5 M EDTA pH 7.4. Afterwards, the samples were dehydrated by a 1-hour wash with increased concentrations of ethanol (70%, 85%, 95% and 2 × 100%), transferred into Histo-Clear (Bar-Naor, Israel) twice for 2 h each and embedded in paraffin blocks. By using a Leica (Agentec, Yakum, Israel) microtome, we prepared 5 μm transverse tissue sections for histological staining [27].

The sections were deparaffinized by heating the slides, followed by two washes with xylen, rehydration by washes with decreasing ethanol concentrations (2 × 100%, 95%, 85%, and 70%) and a final wash with distilled water. Weigert’s iron hematoxylin solution, a fast- green solution and acetic acid were used for Safranin-O staining. The sections were dried and mounted with DPX mounting for histology.

### 2.4. Imaging and Measurement of GPs

Stained transverse sections of tibiae were viewed by an eclipse E400 Nikon light microscopy with ×4, ×10, ×20 or ×40 objectives, using light filters. Images were taken by a high -resolution camera (Olympus DP 71) controlled by Cell A software (Olympus, Japan). The thickness of the entire GPs, the proliferative zone (PZ) and the hypertrophic zone (HZ) as well as the number of cells were measured using the Cell A software with a measuring tool feature. Measurements were performed on these sections from 5 different animals from each group. In each slide, 10 random locations throughout the GPs were selected and measured [28,29]. 

### 2.5. Bone Microarchitecture 

Right femora were scanned using a Skyscan 1174 (Skyscan, Bruker, Belgium) X-ray computed micro-tomography device. Images were obtained at 50 kV X-ray tube voltage and 800 μA current, using a 0.25 mm aluminum filter, 4000 ms exposure time, and 15 μm optical resolution. For each specimen, a series of 900 projection images were obtained (a rotation step of 0.4°, averaging 2 frames, for a total 360° rotation). A stack of 2-D X-ray shadow projections was reconstructed to obtain images using NRecon software (Skyscan). Next, images were subjected to morphometric analysis using CTAn software (CT Analyser 1.13.5.1, Skyscan). Morphometric parameters were calculated as suggested by recent guidelines for bone microstructure assessment. To analyze the diaphyseal cortical region, 200 slices, centered at the mid diaphysis, equivalent to 2.764 mm, were chosen. Global grayscale threshold levels for the cortical region were between 71 and 255. For the trabecular region, a total of 150 slices, equivalent to 2.073 mm of the bone, were selected, and adaptive grayscale threshold levels between 58 and 255 were used. Two phantoms with known density (0.25 and 0.75 g/cm^3^) were scanned under the same conditions of the femora samples allowing to measure the cortical diaphysis BMD (bone mineral density); quantification were carried out using CTAn software [8,30].

The 3rd–5th lumbar vertebrae were scanned and analyzed as well. The spatial resolution was 18 μm and the total rotation was 180°, except that all other parameters were identical to the femoral scans. The region chosen for the analysis was manually selected and consisted of 120 slices of the 5th vertebra starting from the proximal end-plate. Adaptive grayscale threshold levels between 66 and 255 were selected for the analysis of the trabecular vertebral region. 

The length of the femora and 3rd–5th segments of the lumbar vertebrae were measured using the Micro-CT device prior to the scans. By using the Amira software (v.6.4, FEI, Hillsboro, OR, USA) reconstructed scans were volume-rendered to visualize the 3D morphology and BMD variation using visualization, of the selected sample from each group. 

### 2.6. Three Point Bending for Bone Mechanical Analysis 

Right femora were tested using an Instron mechanical tester (Model 3345). Each bone was placed within a custom- built saline containing testing chamber and on two supports having rounded profiles (2 mm in diameter), so that the supports were in touch with the posterior aspect of the diaphysis. The distance between the stationary supports was set to 10 mm, to ensure that the relatively tubular portion of the mid-diaphysis rests on these supports. A pronged loading device was applied to the anterior surface of the bones, precisely in the middle between the two supports. First, an initial preload of 0.1 N was applied to hold the bone in place; following that, the prong was advanced at a constant rate of 600 µm/min, loaded up to the fracture point, identified by a sudden >20% decrease in load [8]. Force-displacement data were collected by the Instron software BlueHill (version 2.0, Instron Corporation, Norwood, Massachusetts, USA) at 10 Hz. The resulting force-displacement curves were used to calculate bone stiffness, bone yield point, load of fracture, maximal load and area under the curve was measured to calculate the total energy to fracture (E to F) [31].

### 2.7. Macro Ad Micro-Nutrient Analysis

Macro nutrient analysis of the three supplements was performed by analytical laboratory AMINOLAB ((Ness Ziona, Israel). Analyses of fatty acids (FAs), amino acids (AAs) and minerals composition were conducted by the Interdepartmental Equipment Unit, The Robert H. Smith Faculty of Agriculture, Food and Environment, The Hebrew University of Jerusalem (Israel). Qualitative analysis of FAs was performed by gas chromatography (GC); liquid chromatography-tandem mass spectrometry (LC-MS/MS) was used for AAs quantitative analysis after hydrolysis of the samples; and last, Inductively Coupled Plasma Optical Emission spectroscopy (ICP-OES) and Inductively Coupled Plasma Mass Spectrometry (ICP-MS) were performed after microwave-assisted nitric acid digestion of the samples in order to assess the mineral content in the supplements. 

### 2.8. Caecum Sample Collection and DNA Extraction

At the day of the sacrifice, caecum from all 48 rats were collected and stored at −80 °C until the day of the extration. 0.25 gr of defrosted caecum content were extracted using QIAamp PowerFecal kit (Qiagen, Hilden, Germany) according to the manufacturer’s instructions. 

### 2.9. 16SrRNASequencing

16SrRNA Sequencing were performed by the University of Illinois at Chicago Core for Research Informatics (UICCRI), using single-end reads with a length of 272 bp with coverage of ~38,500 per sample. Sequencing trimming was done by cutadapt, Amplicon Sequence Variants (ASVs) were inferred using DADA2 and taxonomy was classified against the SILVA 132_16S database. A biom files for each experiment was received. 

### 2.10. Bioinformatics Analysis

Bioinformatics analysis of genus level was conducted by the Bioinformatics Unit, Weizmann Institute of Science (Rehovot, Israel). The threshold for significant differential expression (DE) was set at: padj for multiple testing by Benjamin-Hochberg procedure ≤ 0.05, |log2FoldChange| ≥ 1 and max count ≥ 30 [32]. Principal coordinate analysis (PCoA) was calculated by using Bray-Curtis distances matrix. PCoA is an ordination analysis that simplifies high dimensional data into 2D plot. The axes of the plot are ranked in order of importance: differences along the first principal component axis (PC1) are more important than differences along the second principal component axis (PC2). In order to evaluate the diversity in each of the tested caecum samples Alpha diversity was calculated using the Shannon index. Differential abundance analysis was carried out using Deseq2 on the counts produced by Washington’s lab [33]. 

### 2.11. Statistical Analysis

All data are expressed as mean ± SD. The significance of differences between groups was determined using JMP 14.0.0 Statistical Discovery Software (SAS Institute 2000) by one-way analysis of variance. Differences between groups were further evaluated by a Tukey-Kramer HSD test and considered significant at *p* < 0.05.

## 3. Results

In order to evaluate the influence of unbalanced diet (UBD) and different nutritional supplements (*Spirulina* algae, *Pleurotus eryngii* mushroom and *Agaricus bisporus* mushroom) added to this diet, on skeletal development, a 6-week long experiment was conducted on SD rats after weaning (3 weeks old). This time line is mimicking the human growth period until sexual maturation [8]. 

### 3.1. The Positive Effect of Nutritional Supplements on Growth Pattern 

The effect of the diets on growth was first determined in terms of body weight and length. Body weight represents whole body growth rate, including energy balance, whereas body length is an indicator for longitudinal bone growth. 

The UBD group gained less weight compared to the CON group, while the supplemented groups, UBD + SP, UBD + PE and UBD + AB, gained more weight than the UBD group and managed to partially rescue growth inhibition. The UBD + MIX group showed the highest weight gain from all UBD groups, and by the end of the experiment did not differ from the CON groups (Figure 1A; Appendix A for statistical differences). Body length was measured from the tip of the nose to the end of the tail. Results showed the same patterns as in body weight—UBD rats were shorter than the CON rats, and by the end of the experiment differ also from the UBD supplemented groups. No statistical difference was found between UBD + AB, UBD + MIX and CON groups in the last length measurement (Figure 1B,D; Appendix A for statistical differences). Femur and 3rd–5th lumbar vertebrae length have been measured in the end of the experiment as well. All four supplemented groups demonstrated higher results in these measurements compared with the UBD group (Figure 1E,F). 

Rats from the UBD group consumed fewer calories than all the other. This modification happened subsequent to the slowdown in their growth. Deceleration in growth occurred at the 5th day, while the decline in caloric intake started on day 15th of the experiment (Figure 1C; Appendix A). A slowdown in growth was also observed in the supplemented groups through the same period; however, rats did not reduce their caloric intake which allows them to subsequently improve their growth. Thus, rats which received supplementation consumed more calories than rats consuming UBD, throughout the experiment, and demonstrated growth patterns approximating the CON group (Figure 1; Appendix A).

Since longitudinal bone growth originates from the GP [34], and abnormal impact of the UBD on bone elongation was observed, we performed histological analysis. Safranin-O staining which stains the cartilage in a noticeable red-orange was performed. We found co-responding results with the rats’ length. Both GP thickness and total number of cells were negatively affected by the unbalanced diet and improved thanks to the different supplementations (Figure 2).

### 3.2. The Positive Effect of Nutritional Supplements on Bone Quality

Consumption of a balanced diet is one of the main environmental factors necessary for adequate skeletal development and fulfilment of its full genetic potential [35]. Some of the critical properties encompassed by the term bone quality are trabecular architecture, cortical geometry and mechanical integrity [36]. Micro-CT and three-point bending techniques were used to estimate these parameters. 

Analysis of the femora trabecular bone demonstrated major alterations to bone microarchitecture in rats from the UBD group. Lower values of BV/TV% and Tb.N and higher values of Tb.Sp were found in rats from the UBD group compared with those from the CON group. Twenty-five to thirty-seven percentage of improvement occurred in these parameters in the supplemented groups. Tb.Th that was also negatively affected by the UBD consumption was fully rescued by the supplements to the CON level (Table 4; Figure 3). 

Vertebral bone is for the most part a trabecular bone compartment, and surprisingly, analysis of the 5th vertebra yielded milder effects of the diets (Table 4). These differences could be implied by a different growth pattern between the spine and the limbs, or by different mechanical forces subjected between these two bones.

Cortical bone analysis revealed low values in parameters such as Ct.Ar/Tt.Ar and Cs.Th in the UBD groups which were improved in the supplemented groups. BMD, which reflects the mineralization level of the bones, showed lower values in rats consuming UBD. The UBD + AB and UBD + MIX diets improved the mineral density of the cortical bone the most, reaching to the same level as the CON group (Table 4; Figure 3). The skeleton plays a critical structural role in bearing functional loads, and failure to do so results in fractures; thus, it is valuable to evaluate the mechanical properties of the bones. We found that all tested mechanical parameters were lower in bones of rats fed UBD. The linear slope which represents the stiffness of the bent object and E to F which defined as the work that must be done in order to fracture the bone were lower in the UBD group compared to the CON group, and tended to be improve in the supplemented groups. Moreover, yield point (load at which the load-deflection curve ceases to be linear, indicating micro-damage accumulation in the bone material) and maximal load were lower in the UBD group than in the CON group, and partially improved in the supplemented groups (Table 4).

### 3.3. Diet Composition Analysis of the Supplements

To further comprehend the mechanism by which the supplements commence their beneficial effect, diet composition was investigated [37]. The results in Figure 4 displayed the percentage of carbohydrates, proteins and fats found in the supplements. The main component of *Spirulina* is protein and of mushrooms carbohydrate. Fat fraction in the three supplements is the smallest, and almost undetectable in *Spirulina*.

#### 3.3.1. Treatment of Amino Acids and Mineral Deficiencies 

Amino acids content was evaluated in the three supplements by LC-MS/MS [38]. Except tryptophan, which destructs during hydrolysis all essential AA were found within the nutritional supplements (Appendix A). The AA content in the two mushrooms is quite similar, with some advantage to PE, while greatest amounts were found in SP powder for both essential and non-essential AA (Appendix A). Although the supplements contained all the essential AA, calculation of AA content in kg diet relative to the required minimum presented by the nutritional research council (NRC) [39] (values presented in %), showed poor values in the supplemented groups (Appendix A). When calculation of the essential AA consumed by the rats throughout the experiment was done; higher amounts in the supplemented groups as compared with the UBD group were found, most probably due to their higher food intake. According to the NRC requirements, only in phenylalanine + tyrosine (tyrosine may supply up to 50 percent of phenylalanine and thus calculated together) rats in SP, AB and MIX supplementations groups reached the necessary minimum. Comparison to the CON group still demonstrated major deficiencies (Figure 5A).

Evaluation of the mineral content in SP, PE and AB powders was performed by ICP-OES and ICP-MS techniques [40]. Compared to the mushrooms, spirulina was found being rich specially in sodium, chloride, calcium and magnesium. The highest amount of potassium was found in PE mushroom, while AB was abundant in iron. None lethal trace of lead, tin, cadmium and mercury were found in the three supplements as well (Appendix A). The essential mineral content in kg diet relative to the required minimum presented by the NRC (values presented in %) was calculated and as in AA, none of the dietary supplements have been able to improve the deficiencies caused by the UBD. Levels above the requested minimum according to NRC were in sodium, chloride and zinc, regardless of the added powders (Appendix A). When comparing the amount of total essential minerals consumed by the rats during all 6 weeks of experiment, similar to the amino acids: the supplemented groups consumed more minerals compare with the UBD group, however still could not reach the CON or the NRC levels. Zinc is the only mineral that the supplemented rats consumed more of it compared to both UBD and NRC (Figure 5B).

#### 3.3.2. Functional Fatty Acids

Analysis of the supplement and the main source of fat in the diets, soy oil, were made by GC [41]. The results suggest a resemblance between the soy oil and the two mushrooms, with 18:2 *cis* 9,12 being the main FA followed by 18:1 *cis* 9 and 16:0. Both 16:0 and 18:2 *cis* 9,12 were also found in *Spirulina*; however, in a different ratio, 16:0 was the main FA detected and it represents about 50% of *Spirulina* FA. 18:3 cis 6,9,12 and 16:1 cis 9 were also found in spirulina, while 18:3 cis 6,9,12 being the second most abundant FA (Figure 6). The presence of these FA in the diet, although in minute amounts may generate modifications in the cellular or microbiome level.

### 3.4. Microbiome Analysis

In order to assess the specific changes in the rats’ intestinal microbiota, 16SrRNA sequencing was preformed from caecum samples of 9 weeks old rats.

Alpha diversity was calculated using the Shannon index in order to evaluate the diversity in each of the tested caecum samples. Significantly decreased was found in the UBD group compared with the CON group. Supplementation of PE, AB or Mix was able to significantly increase the alpha diversity; however only trends of improvement were found with SP supplementation (Figure 7A).

Principal coordinate analysis (PCoA) was used with the aim to evaluate the ecological distance between samples and produce a graphical configuration in a low-dimension. Figure 7B presents PCoA plot based on the Bray-Curtis distances matrix, which was calculated on the genus level. The calculated distances in the CON samples was much smaller; the group is mostly clustered together and separated from the other experimental groups. All four supplementation groups led to a smaller variance in the microbiome composition inside the group and between the groups as well. Samples from these groups are more clustered together compared with the UBD group but not as in the CON and separation between the supplemented groups and the CON group was still observed. 

On an effort to point out the bacteria that might take part in mediating the effect of UBD consumption on the skeletal system we focused on the significantly bacteria changes in the UBD compared with the CON group and examined the influence the different supplements had on the abundance of these bacteria. Genera abundance was first assessed using a heat map (Figure 8A). 

The genera *Mucispirillum* and *Lachnospiraceae_NK4A136_group* differed significantly between the UBD and the CON groups: increased and decreased respectively with UBD consumption. We have not seen any changes in these bacteria in the supplemented groups compare to both the UBD and the CON groups.

We saw an opposite effect in the UBD and the supplemented groups in five different bacteria: *Butyricimonas, Parabacteroides, Rhodospirillales /other, Parasutterella* and *Escherichia /Shigella* which were elevated in the UBD compared with the CON group, and were reduced in the supplemented groups compared with the UBD. In addition reduced levels of *Acetatifactor* were detected in the UBD compared with the CON group and the supplements managed to improve this reduction (Figure 8B). Supplementation of SP, PE and AB to the UBD decreased the abundance of *Butyricimonas* and *Rhodospirillales/other* in a significant way compared to the UBD without supplements, and no statistical difference compared to the CON group was observed (Figure 8B). We also saw reduction in *Parabacteroides* and *Parasutterella* bacteria compared to the UBD while administration of the different supplements was performed. The abundance of these two bacteria was even higher compared to the CON group with PE and AB mushrooms addition to the diet (Figure 8B). All four supplements elevated the levels of *Acetatifactor* compared to the UBD alone, and MIX supplementation was the only one that cause to a significant difference compared to the CON group as well (Figure 8B). Tendency to higher abundance of *Escherichia /Shigella* in the UBD compare to the CON group was found and supplementation with SP, PE and MIX lowered this effect in a significant way (Figure 8B). These results suggest that some of the supplements were able to equalize the abundance of different bacteria to CON level, while others had a partial effect. 

## 4. Discussion

In this research we establish a model that simulates the massive changes in the general population food habits and mimics the diet prevalent in both developed and developing countries. We shed a light on the importance of balanced nutrition, and revealed that consumption of unbalanced diet lead to severe growth retardation, disrupted bone structural parameters, decreased bone mineral density (BMD) and reduced strength and rigidity. Low doses of botanical dietary supplements were able to rescue the damage caused by this diet; leading to partial or full rescue to the control levels.

To elucidate the foundation for the beneficial effect the supplements have on the rats’ growth and skeletal quality, several explanations were tested. First, the nutritional composition of the supplements was examined to see if it compensates as for the deficiencies in the UBD. Despite the abundant ingredients in algae and the mushrooms, their addition to the diet did not quantitatively compensate for the huge deficiencies. Due to the minute amount added to the UBD diet (as recommended by manufacturers) the supplements failed to increase the amount of essential amino acids and minerals in the diet. Even when the increased food consumption of the supplemented groups was taken into account the amounts of essential nutrients did not reach the minimal requirements according to the NRC [39]. The dietary supplements do not improve the rats’ growth and skeletal quality by reversing the shortages of the unbalanced diet; suggesting they function as growth enhancers rather than nutritional improvers.

Next we examined whether the supplements effect might be related to their impact on the GM. Rise attention had been recently given to the effects microbial communities have on the human body and its overall health; and to the negative health consequences associated with microbial imbalance [13]. This imbalance could be attributed to the gain or loss of community members, known as dysbiosis. Dysbiosis, is commonly associated with impaired gut barrier function, increased pathogen translocation and inflammatory cell activation, and has been directly linked to several pathological conditions such as: inflammatory bowel disease (IBD), irritable bowel syndrome (IBS), diabetes, obesity and cancer [42,43]. High fat diets, sugar rich-diet and high red meat consumption are examples of diets already found to induce dysbiosis [44]. Bone metabolism is tightly regulated by the immune system, both locally and systemically. Macrophage, dendritic cells and T-helper cell (Th) subpopulations, produce pro-inflammatory cytokines such as TNFα, TGFβ, Type I interferon (IFNγ), IL-6, 17 and 22, known to regulate bone growth, and promotes resorption over formation [45]. 

Moreover, metabolic damage was found in dysbiosis due to reduced vitamins biosynthesis, impaired nutrient absorption and enzymatic breakdown of compounds as polysaccharides and polyphenols. The colon receives indigestible carbohydrates and proteins that represent 10–30% of the total ingested energy. Without the activity of the colonic microbiota these nutrients would generally be eliminated via the stool [46]. Short chain fatty acids (SCFA) are the most investigated guts products, that have been linked to T cells regulation [47], improved calcium absorption [48] and IGF-1 secretion, a dominant bone growth factor [49]. Indole, fermentation product of aromatic amino acid tryptophan, stimulates the production of mucin and the proliferation of goblet cells, to maintain intestinal impermeability [50]. It is well established that vitamins from B group and vitamin K can be synthesized by the GM and be used by the host [51]. Thus, decreased bacterial diversity, as found in the UBD group may act negatively on the skeleton. 

When focusing on the specific changes occurring in the bacteria’s levels following administration of an unbalanced diet and the supplements, we found several bacteria associated with inflammation to be higher in the UBD group. For instance, *Mucispirillum* which is associated with active colitis [52]. *Actinobacteria* and *Proteobacteria* phylum are associated with IBD. *Escherichia/Shigella*, genus from the *Proteobacteria* phylum is one of the main bacteria associated with Crohn’s disease [43] and *Parasutterella*, a relatively new genus also part of the *Proteobacteria* phylum with advanced IBD [53]. Higher levels of these bacteria were found in the UBD group and reduction occurred in the group consuming the supplements. Up to 50% of IBD patients experience at least one extra-intestinal manifestation; among them is the involvement of reduced bone density [54]. It appears that alerted bone profile in IBD can be caused by number of reasons: inflammatory process in the gut with high levels of cytokines, nutritional alterations and nutrients deficiency due to inadequate diet intake and/or malabsorption [55].

Higher levels of *Parabacteroides* species, that were elevated by consumption of UBD in our experiments, were found in osteoporosis and osteopenia patients [56]. Osteopenia and osteoporosis are two bone diseases often associated with ageing. Although we used young rats in our experiments, their bone profile and skeletal characterization were comparable to those suffering from osteoporosis; reduced skeletal strength and BMD along with microstructural defects. Interestingly, SP, PE and MIX supplementation were able to decrease the elevated levels of Parabacteroides in the rats’ caecum.

Decreased levels of *Lachnospiraceae_NK4A136_group* and *Acetatifactor* were detected in the UBD group. Previous studies have indicated that *Lachnospiraceae _NK4A136_group* may have an anti-inflammatory effect [57]. It is positively correlated with acetic and butyric acid, two forms of SCFA, SOD, GSH-Px and GSH [58]. *Acetatifactor* metabolite products can activate the G protein-coupled bile acids (BAs) receptor (TGR5) signaling, which has a few beneficial functions in the skeletal system [59]. First, TGR5 can promote the secretion of glucagon-like peptide-1 (GLP-1) [59,60], that was recently associated with bone tissue. GLP-1 receptor (GLP-1 R) has been found on immature osteoblastic cell lines, on mouse osteoblast cells and in bone marrow stem cells [61]. It was found to increase the number of osteoblasts on the surface of trabecular bone and up regulate the expression of Runx2, ALP, collagen type 1, and osteocalcin [62]. GLP-1 may also promote bone formation through the reduction of sclerostin expression, which results in increased Wnt signaling pathway and bone formation [61,62]. Suppression of bone resorption by GLP-1 is associated with the expression of GLP1R in thyroid C cells where it can promote the secretion of calcitonin, an inhibitor of bone resorption by osteoclasts [63]. TGR5 signaling can also increase the activity of type 2 deiodinase (D2) which converts thyroxine (T4) into the active form Triiodothyronine (T3) [59]. T3 acts in chondrocytes and osteoblasts to regulate intramembranous and edonchondral ossification and control the rate of linear growth and bone maturation. Studies suggest that T3 regulates a number of key growth factor signaling pathways including IGF-1, FGF, PTHrP, Ihh and Wnt to influence skeletal growth [64]. In all four supplemented groups, improvement in Acetatifactor abundance was observed. Hormonal testes we preformed reveled significant decrease in T4 in the UBD group, which improved thanks to the supplements. No differences were found in TSH, T3 and IGF-1 (data not shown).

Lastly, we evaluated the active ingredients or fermentation products found in the supplements. Spirulina contains several active ingredients as phycocyanin, β-carotene, γ-linolenic fatty acid and omega-3 fatty acids; all of which show antioxidant, immunomodulatory and anti-inflammatory activities [24,65]. Spirulina also contains polysaccharides, resistant starch and oligosaccharides which known as dietary fibers. Humans lack the ability to digest β 1→4 linkages so the undigested materials reach the large intestine and undergo fermentation to SCFA, which has a variety of health beneficial as described earlier [65]. 

The bioactive substances found in mushrooms are mainly divided into: β-glucans, types of polysaccharides, and secondary metabolites as polyphenols [20]. Current literature suggests that long-term consumption of diets rich in polyphenols protects against osteoporosis and play a role in bone metabolism. They found to be protective against oxidative stress and capable of down regulating osteoclast differentiation and activity by inhibition of RANKL and COX-2 production and stimulation of OPG [66]. Das, Mukherjee, and Mitra et al. showed that black tea extract prevents bone loss induced by ovariectomy (OVX) in rats. The OVX group had significantly higher serum ALP, TRAP and lower bone density, which all were improved with the supplementation [67]. Bone regeneration effects of β-glucans as well as its anti-osteoporotic properties have been studied using in vitro and animal experiments. Increased BMD and calcium bioavailability were observed in rat model for osteoporosis. Human-based clinical trials are limited and reveled beneficial effects when added to healthy menopausal women: significant changes in serum bone-specific ALP, serum osteocalcin, urinary cross-linked N-telopeptide of type-1 collagen, urinary cross-linked C-telopeptide of type-1 collagen, calcium, and phosphorus levels [68,69].

In this study, we created for the first time a model that simulates the massive changes in the general population food habits: higher consumption of fat and insufficient amount of protein, vitamins and minerals. We investigated the diet effect as intact and not as separate components; the primary advantage of this approach is the translational potential from model organisms to humans.

Dramatic effect to the bone was observed along with slowdown in the rats’ growth rate. Supplementation of the deficient diet with botanical supplements showed surprising results: significant improvement to the morphological and mechanical bone parameters and growth patterns similar to those of the control group. Since we show that the mushrooms and spirulina were unable to correct the deficiencies of the diet, we suggest that the major positive effect of the supplements were attributed to their effect on the microbiome or thanks to their active ingredients and fermentation products.

## Figures and Tables

**Figure 1 nutrients-13-04316-f001:**
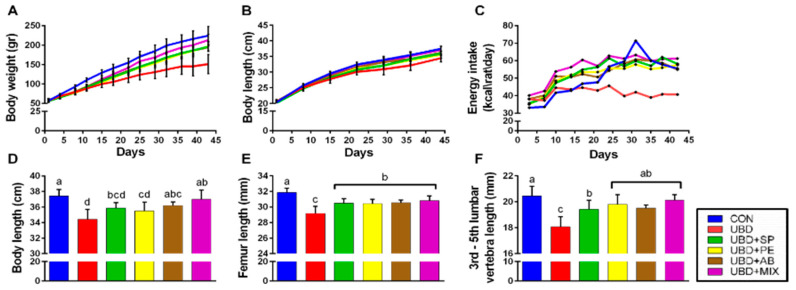
Growth patterns and food consumption. (**A**) Body weight during the entire experiment (g). (**B**) Body length during the entire experiment (cm). (**C**) Energy intake (kcal/rat/day). (**D**) Body length in the end of the experiment (cm). (**E**) Femur length measured by micro-CT (mm). (**F**) 3rd–5th lumbar vertebra length measured by micro-CT (mm). Values are expressed as mean ± SD of *n* = 8 rats/group, different superscript letters are significantly different (*p* < 0.05) by one-way ANOVA followed by Tukey’s test.

**Figure 2 nutrients-13-04316-f002:**
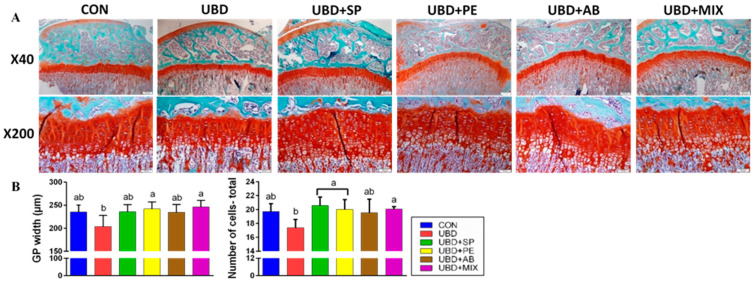
Histological evaluation of the tibial growth plates (GPs) of 9 weeks old rats. Transverse tissue sections of 5 μm were prepared by microtome. GP width and number of cells has been measured in cell A software at 10 selected locations. (**A**) Safranin-O staining of the tibial GP. (**B**) GP width (µm); Number of cells. Values are expressed as mean ± SD of *n* = 5 rats/group, different superscript letters are significantly different (*p* < 0.05) by one-way ANOVA followed by Tukey’s test.

**Figure 3 nutrients-13-04316-f003:**
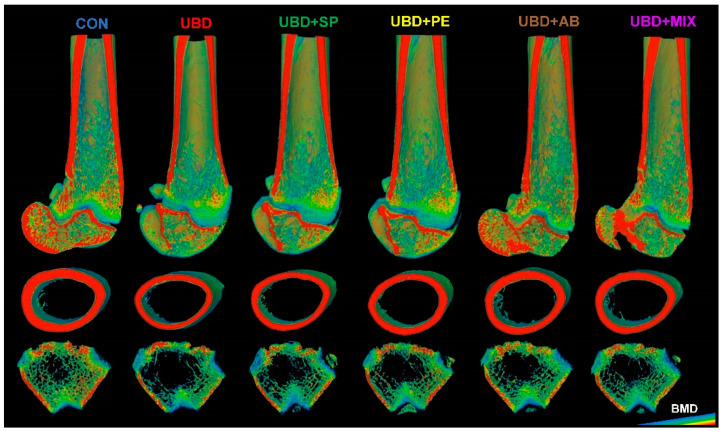
Three-dimensional reconstructions of femora scaled for mineral density using AMIRA software. Sagittal sections of the distal femora bone; Transverse sections of cortical area at mid-diaphysis; Transverse sections of trabecular bones at the distal epiphysis above the growth plate.

**Figure 4 nutrients-13-04316-f004:**
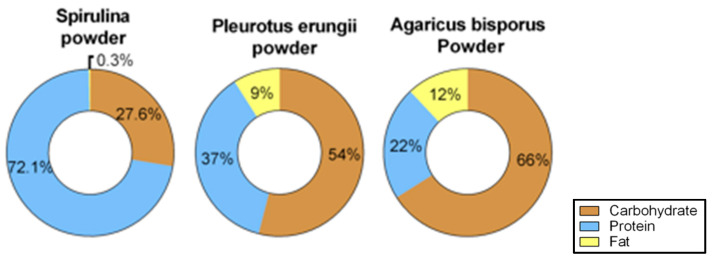
Macronutrient’s composition. Caloric distribution percentage of macronutrients in the three supplements.

**Figure 5 nutrients-13-04316-f005:**
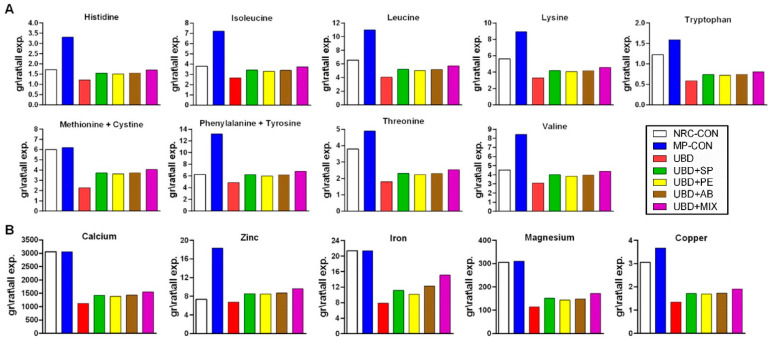
Essential amino acids and minerals consumption. (**A**) Grams of essential amino acids consumed by the rats during the entire experiment. (**B**) Grams of essential minerals for skeletal development consumed by the rats during the entire experiment. The NRC-CON group is based on the CON food intake and the NRC requires recommendations.

**Figure 6 nutrients-13-04316-f006:**
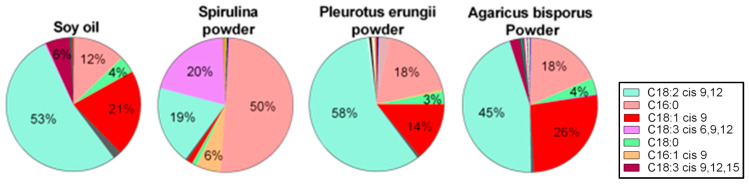
Fatty acid profiles. Lipids were analyzed by gas chromatography–mass spectroscopy.

**Figure 7 nutrients-13-04316-f007:**
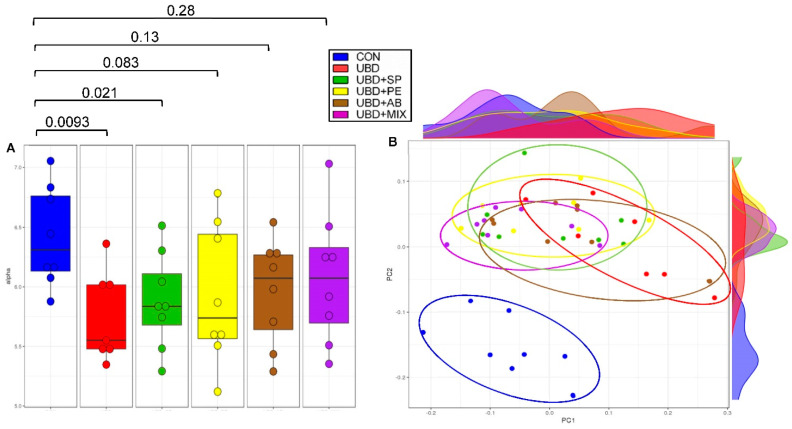
Bacterial diversity between and within samples. (**A**) Alpha diversity was calculated using Shannon index. (**B**) Principal coordinates analysis was calculated based on bray-curtis distances matrix. *p*-value was calculated using Pairwise comparisons Wilcox. Test.

**Figure 8 nutrients-13-04316-f008:**
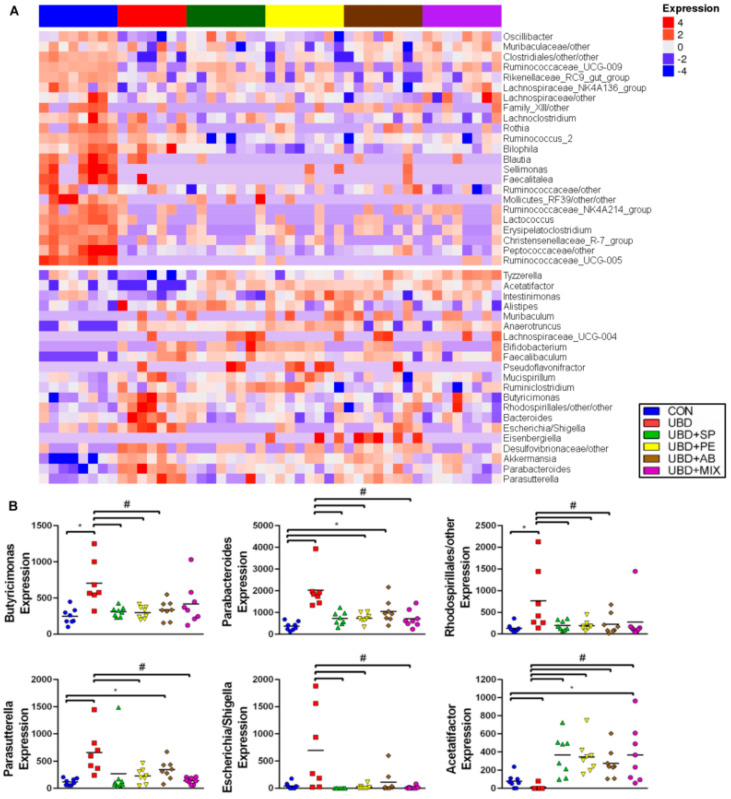
Change in the bacterial profile. (**A**) Results were analyzed using Deseq2 and assessed using heatmap. (**B**) Differential abundance analysis was carried out using Deseq2 on the counts products. *p* value was adjusted for multiple testing by Benjamin-Hochberg procedure (Padj < 0.05). (*) Significant difference compared with the CON group. (#) Significant difference compared with the UBD group.

**Table 1 nutrients-13-04316-t001:** Macronutrient, vitamin and mineral composition of the experimental diets. Values are presented as gram per dry weight.

	Control Diet	Unbalanced Diet
Ingredient	g\kg Diet	g\kg Diet
Cornstarch	397	438.7
Casein (≥85% protein)	200	105
Dextrinized cornstarch (90–94% tetrasaccharides)	132	145.7
Sucrose	100	110
Soybean oil	70	118.8
Fiber	50	54
Mineral mix (AIN-93G-MX)	35	18.9
Vitamin mix (AIN-93-VX)	10	5.4
L-Cystine	3	1.6
Choline bitartrate (41.1% choline)	2.5	1.34
Tert-butylhydroquinone	0.014	0.015

**Table 2 nutrients-13-04316-t002:** Caloric composition of the experimental diets. Values are presented as calories per weight with added water.

	Control Diet		Unbalanced Diet	
Ingredient	Kcal\kg Diet	% Kcal	Kcal\kg Diet	% Kcal
Cornstarch	1402	63.5	1559.4	65
Dextrinized cornstarch (90–94% tetrasaccharides)	465.5		518	
Sucrose	352.7		392	
Casein (≥85% protein)	705	20.5	374	10
L-Cystine	10.6		5.6	
Soybean oil	555.5	16	949.9	25
Total calories	3491.3		3799	
Kcal/gr	3.49		3.799	

**Table 3 nutrients-13-04316-t003:** Supplements added to the diet. Values are presented as gram per dry weight and as Kcal per dry weight of the supplements added to kg diet.

Ingredient	g\kg Diet	Kcal\kg Diet
Spirulina algae	5	19
Pleurotus eryngii	3.5	15.765
Agaricus bisporus	3.5	13.618

**Table 4 nutrients-13-04316-t004:** Morphometric and mechanic characteristics.

	CON	UBD	UBD + SP	UBD + PE	UBD + AB	UBD + MIX
Femora trabecular bone microarchitecture						
BV/TV (%)	38.222 ± 2.81 ^a^	23.261 ± 2.79 ^c^	30.44 ± 3.49 ^b^	30.263 ± 2.99 ^b^	33.214 ± 1.29 ^b^	33.697 ± 3.19 ^b^
Tb.Th (mm)	0.119 ± 0.009 ^a^	0.105 ± 0.005 ^b^	0.118 ± 0.007 ^a^	0.117 ± 0.006 ^a^	0.117 ± 0.04 ^a^	0.121 ± 0.008 ^a^
Tb.N (1/mm)	3.203 ± 0.15 ^a^	2.205 ± 0.19 ^c^	2.583 ± 0.2 ^b^	2.592 ± 0.19 ^b^	2.841 ± 0.16 ^b^	2.771 ± 0.16 ^b^
Tb.Sp (mm)	0.264 ± 0.06 ^c^	0.572 ± 0.12 ^a^	0.466 ± 0.06 ^ab^	0.45 ± 0.05 ^b^	0.398 ± 0.07 ^b^	0.378 ± 0.03 ^b^
5th vertebra trabecular bone microarchitecture						
BV/TV (%)	36.797 ± 1.4 ^a^	30.568 ± 1.24 ^c^	32.841 ± 1.77 ^bc^	33.146 ± 2.41 ^bc^	35.011 ± 2.15 ^ab^	31.881 ± 2.57 ^c^
Tb.Th (mm)	0.128 ± 0.005 ^a^	0.111 ± 0.003 ^d^	0.116 ± 0.003 ^c^	0.115 ± 0.003 ^cd^	0.121 ± 0.003 ^b^	0.115 ± 0.003 ^cd^
Tb.N (1/mm)	2.871 ± 0.09 ^a^	2.762 ± 0.1 ^a^	2.84 ± 0.12 ^a^	2.869 ± 0.17 ^a^	2.901 ± 0.15 ^a^	2.775 ± 0.19 ^a^
Tb.Sp (mm)	0.297 ± 0.017 ^a^	0.328 ± 0.017 ^a^	0.312 ± 0.012 ^a^	0.307 ± 0.025 ^a^	0.313 ± 0.03 ^a^	0.322 ± 0.037 ^a^
Femora cortical bone microarchitecture						
Ct.Ar/Tt.Ar (%)	48.925 ± 1.69 ^a^	40.378 ± 1.87 ^c^	43.534 ± 1.06 ^b^	42.945 ± 1.69 ^b^	44.6 ± 1.38 ^b^	44.565 ± 1.61 ^b^
Cs.Th (mm)	0.466 ± 0.02 ^a^	0.343 ± 0.02 ^c^	0.384 ± 0.01 ^b^	0.385 ± 0.02 ^b^	0.387 ± 0.01 ^b^	0.392 ± 0.01 ^b^
Ma.Ar (mm^2^)	5.324 ± 0.4 ^a^	5.108 ± 0.59 ^a^	5.081 ± 0.25 ^a^	5.299 ± 0.35 ^a^	4.828 ± 0.3 ^a^	4.981 ± 0.39 ^a^
BMD (g/cm^3^)	1.26 ± 0.03 ^a^	1.184 ± 0.01 ^c^	1.205 ± 0.02 ^bc^	1.199 ± 0.02 ^bc^	1.229 ± 0.03 ^ab^	1.23 ± 0.04 a^b^
Femora bone mechanical properties						
Stiffness (N/mm)	268.72 ± 45.5 ^a^	195.6 ± 43 ^b^	225.32 ± 22.9 ^ab^	197.34 ± 44 ^b^	228.49 ± 33.9 ^ab^	202.57 ± 39.1 ^b^
Yield point (N)	45.55 ± 3.9 ^a^	26.31 ± 4.2 ^c^	36.01 ± 3.7 ^b^	34.64 ± 2.7 ^b^	37.8 ± 2.6 ^b^	37.28 ± 3 ^b^
Fracture load (N)	74.13 ± 7.5 ^a^	42.69 ± 11.4 ^b^	52.06 ± 6.2 ^b^	41.81 ± 9.3 ^b^	51.25 ± 6.5 ^b^	50.21 ± 11.2 ^b^
Max load (N)	90.47 ± 7.7 ^a^	63.04 ± 9.3 ^c^	75.32 ± 6.7 ^b^	68.81 ± 5.7 ^bc^	70.01 ± 5.9 ^bc^	73.26 ± 6.2 ^bc^
E to F (N × mm)	83.01 ± 16.5 ^a^	54.93 ± 10.6 ^c^	71.8 ± 11.8 ^abc^	72.04 ± 11.6 ^abc^	63.2 ± 8.7 ^bc^	74.84 ± 17.5 ^ab^

Bones were scanned by Micro-CT to determine geometric parameters. After reconstructing 2D and 3D analyses were performed. Trabecular bone parameters: Bone volume over total volume, BV/TV (%); Trabecular thickness, Tb.Th (mm); Trabecular number, Tb.N (1/mm); Trabecular separation, Tb.Sp (mm). Cortical bone parameters: Cortical area fraction, Ct.Ar/Tt.Ar (%); Cortical thickness, Cs.Th (mm); Medullary area, Ma.Ar (mm^2^); Bone mineral density, BMD (mm^3^). Mechanical properties were evaluated using three-point bending experiment. Biomechanical parameters obtained from load–displacement curve: Stiffness (N/mm); Yield point (N); Fracture load (N); Max load (N); Energy to fracture, E to F (N × mm). Values are expressed as mean ± SD of *n* = 8 rats/group, different superscript letters are significantly different (*p* < 0.05) by one-way ANOVA followed by Tukey’s test.

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
