# Peer review of "The Use of Mushrooms and Spirulina Algae as Supplements to Prevent Growth Inhibition in a Pre-Clinical Model for an Unbalanced Diet"

_nutrients, 2021, doi:10.3390/nu13124316_

Round 1

Reviewer 1 Report

Broad comments

In general, I have no issues with the article, methodology, data analysis and results have correctly been conducted and discussed. Moreover, introduction is adequate and easy to read.

Specific comments

Line 2: Title, I propose to delete the word “medicinal”, since no studies were conducted on the medicinal effects of mushrooms (i.e. pharmacological activity).

Line 69:  join the “s” to their corresponding word:  “supplements”  

Line 78, 79:  scientific names must be highlighted, for example in italics.

All over the document, including tables, figures and supplementary material, scientific names must be highlighted.

Line 93: If possible, you should add the species of the spirulina, just as the scientific names of mushrooms are presented.

Line 123: You should specified the quantities or proportions added of each supplement in the UBD+MIX diet (How much of SP, PE and AB).

Line 142: Change the letter “t” to “r”  in the word pet

Table 3. I suggested to add the quantities or proportions of each supplement in the UBD+MIX diet (How much of SP, PE and AB), as mentioned above.

Line 148: delete (Olympus), it was already mentioned above.

Line 195: for macro and micro-nutrients section, you should add the references of the methods used to determine protein, carbohydrate, fat, amino acids and minerals.

Line 258: I suggest deleting the sentence “consumed less food” since no measure of how much food the rats consumed was presented. Unless it has been measured, then you should specify how it was measured and how much food they consumed.

Line 260: change “between the 10th-15th “ by “started on day 15th “, since in the results it is observed that on day 10th energy intake was even higher than CON.

Line 261: (Figure 1C; supplementary Figure 1) could be removed because is used in line 264 and followed the same idea.

Line 278: I suggest adding a brief qualitative description of the histological photographs. Right after Safranin-O staining statement...

Line 281: The titles of the histological photos must be completed.

Line 289 to 292: This paragraph (up to citation 36) should be placed in the discussion where appropriate.

Line 302 to 303: This paragraph should be placed in the discussion where appropriate.

Line 309 to 311: This statement (up to properties of bones) should be placed in the discussion where appropriate.

Line 320: The numbers in table 4 should be fixed, check the number of digits after the point.

Line 322: It says “table” but it is not understood what it refers to, it should be corrected.

Line 337: on the subtitle, I suggest specifying that the analysis was done on the supplements

Line 347: The subtitle is not well understood, it should be more explicit.

Line 359: “only in phenylalanine + tyrosine rats” the sentence it was not clear and should be detailed.

Line 368: change “N” letter to lowercase “none”

Line 392: the words “extracted and concentrated” must be removed since GC-MS did not perform the extraction and concentration.

Line 405: chance “was” per were, samples were…

Line 425: the six different bacteria were not mention, the bacteria must be mentioned before the subordinating conjunction  “while”

Line 564: you should mention if the amounts of metals (lead, tin, mercury, etc.) found in the supplements are dangerous.

Supplementary material, the titles of tables 3 and 4 are incomplete, they should be completed.

Author Response

We would like to thank the reviewer for his positive attitude to our paper and for the constructive suggestions. 

we feel that the paper is improved, and hope you will find it suitable for publication.

The following modifications has been adopted.

Line 2: Title, I propose to delete the word “medicinal”, since no studies were conducted on the medicinal effects of mushrooms (i.e. pharmacological activity). Done

Line 69:  join the “s” to their corresponding word:  “supplements” Done

Line 78, 79:  scientific names must be highlighted, for example in italics. Done

All over the document, including tables, figures and supplementary material, scientific names must be highlighted. Done

Line 93: If possible, you should add the species of the spirulina, just as the scientific names of mushrooms are presented. Unfortunately this is not possible

Line 123: You should specified the quantities or proportions added of each supplement in the UBD+MIX diet (How much of SP, PE and AB).Done

Line 142: Change the letter “t” to “r”  in the word pet.Done

Table 3. I suggested to add the quantities or proportions of each supplement in the UBD+MIX diet (How much of SP, PE and AB), as mentioned above.This is mentioned in the table and in the text.

Line 148: delete (Olympus), it was already mentioned above. Done

Line 195: for macro and micro-nutrients section, you should add the references of the methods used to determine protein, carbohydrate, fat, amino acids and minerals.Done

Line 258: I suggest deleting the sentence “consumed less food” since no measure of how much food the rats consumed was presented. Unless it has been measured, then you should specify how it was measured and how much food they consumed.Done

Line 260: change “between the 10th-15th “ by “started on day 15th “, since in the results it is observed that on day 10th energy intake was even higher than CON.Done

Line 261: (Figure 1C; supplementary Figure 1) could be removed because is used in line 264 and followed the same idea.Done

Line 278: I suggest adding a brief qualitative description of the histological photographs. Right after Safranin-O staining statement..Done

Line 281: The titles of the histological photos must be completed. Done

Line 289 to 292: Line 302 to 303: Line 309 to 311: we feel that these parts are connections between the parts of the research and not a discussion.

Line 320: The numbers in table 4 should be fixed, check the number of digits after the point. Done

Line 322: It says “table” but it is not understood what it refers to, it should be corrected. Done

Line 337: on the subtitle, I suggest specifying that the analysis was done on the supplements. Done

Line 347: The subtitle is not well understood, it should be more explicit.Done

Line 359: “only in phenylalanine + tyrosine rats” the sentence it was not clear and should be detailed. Done

Line 368: change “N” letter to lowercase “none” Done

Line 392: the words “extracted and concentrated” must be removed since GC-MS did not perform the extraction and concentration.Done

Line 405: chance “was” per were, samples were....Done

Line 425: the six different bacteria were not mention, the bacteria must be mentioned before the subordinating conjunction  “while" Done

Line 564: you should mention if the amounts of metals (lead, tin, mercury, etc.) found in the supplements are dangerous.Done

Supplementary material, the titles of tables 3 and 4 are incomplete, they should be completed.Done

Reviewer 2 Report

This manuscript introduced a model that mimics the modern UBD and tested its effects on longitudinal bone growth, trabecular and cortical bone micro-architecture and bone mineral density in young rats. The authors also compared the impact of dietary supplements, Spirulina algae, Pleurotus Eryngii and Agaricus Bisporus mushroomsin the UBD. The results suggested these botanical dietary supplements have positive effect on the morphological and mechanical bone parameters.

This manuscript establish an important sight in this field and could have significant meaning if published. However, some points should be addressed before accepting for publication in Nutrients.

  1. The quality of some figures should be improved. For example, the y axis of Figure 1B should be changed so that lines were easy to read; the labels in Figure 2A were not complete and should be aligned; the color code of Figure 3 was not complete; etc.

  1. Please provide original data and explain why the UBD group has 7 mice while others have eight for Figure 7 and 8B. Does it mean other figures also used 7 mice for the UBD group? Please show data points for other figures (Figure 1, 2B, and 5).

Author Response

We would like to thank the reviewer for his attitude and suggestions, we have modified the paper according to your requests, and hope you will find it now suitable for publication.

  1. The quality of some figures should be improved. For example, the y axis of Figure 1B should be changed so that lines were easy to read; the labels in Figure 2A were not complete and should be aligned; the color code of Figure 3 was not complete; etc. All figures quality were checked and improved.

  1. Please provide original data and explain why the UBD group has 7 mice while others have eight for Figure 7 and 8B. Does it mean other figures also used 7 mice for the UBD group?  One of the cecum samples (that were sent for outsourcing 16S seq) did not yield a qualitative results, probably due to technical issues either in the DNA preparation or in the sequencing process. in consultant with the bioinformatic department we used all the other samples, since quality and differences were significant even with the lower number in this group. In all other analyses 8 rats were used.
